# Learning from Complementary Labels

**Takashi Ishida[1,2,3]    Gang Niu[2,3]    Weihua Hu[2,3]    Masashi Sugiyama[3,2]**
[1] Sumitomo Mitsui Asset Management, Tokyo, Japan
[2] The University of Tokyo, Tokyo, Japan
[3] RIKEN, Tokyo, Japan
{ishida@ms., gang@ms., hu@ms., sugi@}k.u-tokyo.ac.jp

## Abstract

Collecting labeled data is costly and thus a critical bottleneck in real-world classification tasks. To mitigate this problem, we propose a novel setting, namely *learning from complementary labels* for multi-class classification. A complementary label specifies a class that a pattern does *not* belong to. Collecting complementary labels would be less laborious than collecting ordinary labels, since users do not have to carefully choose the correct class from a long list of candidate classes. However, complementary labels are less informative than ordinary labels and thus a suitable approach is needed to better learn from them. In this paper, we show that an *unbiased estimator* to the *classification risk* can be obtained only from complementarily labeled data, if a loss function satisfies a particular symmetric condition. We derive *estimation error bounds* for the proposed method and prove that the *optimal parametric convergence rate* is achieved. We further show that learning from complementary labels can be easily combined with *learning from ordinary labels* (i.e., ordinary supervised learning), providing a highly practical implementation of the proposed method. Finally, we experimentally demonstrate the usefulness of the proposed methods.

## 1   Introduction

In ordinary supervised classification problems, each training pattern is equipped with a label which specifies the class the pattern belongs to. Although supervised classifier training is effective, labeling training patterns is often expensive and takes a lot of time. For this reason, learning from less expensive data has been extensively studied in the last decades, including but not limited to, semi-supervised learning [4, 38, 37, 13, 1, 21, 27, 20, 35, 16, 18], learning from pairwise/triple-wise constraints [34, 12, 6, 33, 25], and positive-unlabeled learning [7, 11, 32, 2, 8, 9, 26, 17].

In this paper, we consider another weakly supervised classification scenario with less expensive data: instead of any ordinary class label, only a *complementary label* which specifies a class that the pattern does *not* belong to is available. If the number of classes is large, choosing the correct class label from many candidate classes is laborious, while choosing one of the incorrect class labels would be much easier and thus less costly. In the binary classification setup, learning with complementary labels is equivalent to learning with ordinary labels, because complementary label 1 (i.e., not class 1) immediately means ordinary label 2. On the other hand, in $K$-class classification for $K > 2$, complementary labels are less informative than ordinary labels because complementary label 1 only means either of the ordinary labels $2, 3, \ldots, K$.

The complementary classification problem may be solved by the method of learning from *partial labels* [5], where multiple candidate class labels are provided to each training pattern—complementary label $\overline{y}$ can be regarded as an extreme case of partial labels given to all $K-1$ classes other than class $\overline{y}$. Another possibility to solve the complementary classification problem is to consider a multi-label

setup [3], where each pattern can belong to multiple classes—complementary label $\overline{y}$ is translated into a negative label for class $\overline{y}$ and positive labels for the other $K-1$ classes.

Our contribution in this paper is to give a direct risk minimization framework for the complementary classification problem. More specifically, we consider a *complementary loss* that incurs a large loss if a predicted complementary label is not correct. We then show that the classification risk can be empirically estimated in an unbiased fashion if the complementary loss satisfies a certain symmetric condition—the sigmoid loss and the ramp loss (see Figure 1) are shown to satisfy this symmetric condition. Theoretically, we establish estimation error bounds for the proposed method, showing that learning from complementary labels is also consistent; the order of these bounds achieves the optimal parametric rate $\mathcal{O}_p(1/\sqrt{n})$, where $\mathcal{O}_p$ denotes the order in probability and $n$ is the number of complementarily labeled data.

We further show that our proposed complementary classification can be easily combined with ordinary classification, providing a highly data-efficient classification method. This combination method is particularly useful, e.g., when labels are collected through crowdsourcing [14]: Usually, crowd-workers are asked to give a label to a pattern by selecting the correct class from the list of all candidate classes. This process is highly time-consuming when the number of classes is large. We may instead choose one of the classes randomly and ask crowdworkers whether a pattern belongs to the chosen class or not. Such a yes/no question can be much easier and quicker to be answered than selecting the correct class out of a long list of candidates. Then the pattern is treated as ordinarily labeled if the answer is yes; otherwise, the pattern is regarded as complementarily labeled.

Finally, we demonstrate the practical usefulness of the proposed methods through experiments.

## 2  Review of ordinary multi-class classification

Suppose that $d$-dimensional pattern $\boldsymbol{x} \in \mathbb{R}^d$ and its class label $y \in \{1, \ldots, K\}$ are sampled independently from an unknown probability distribution with density $p(\boldsymbol{x}, y)$. The goal of ordinary multi-class classification is to learn a classifier $f(\boldsymbol{x}) : \mathbb{R}^d \to \{1, \ldots, K\}$ that minimizes the classification risk with multi-class loss $\mathcal{L}\big(f(\boldsymbol{x}), y\big)$:

$$R(f) = \mathbb{E}_{p(\boldsymbol{x}, y)}\big[\mathcal{L}\big(f(\boldsymbol{x}), y\big)\big], \tag{1}$$

where $\mathbb{E}$ denotes the expectation. Typically, a classifier $f(\boldsymbol{x})$ is assumed to take the following form:

$$f(\boldsymbol{x}) = \underset{y \in \{1, \ldots, K\}}{\arg\max}\ g_y(\boldsymbol{x}), \tag{2}$$

where $g_y(\boldsymbol{x}) : \mathbb{R}^d \to \mathbb{R}$ is a binary classifier for class $y$ versus the rest. Then, together with a binary loss $\ell(z) : \mathbb{R} \to \mathbb{R}$ that incurs a large loss for small $z$, the *one-versus-all* (OVA) loss[1] or the *pairwise-comparison* (PC) loss defined as follows are used as the multi-class loss [36]:

$$\mathcal{L}_{\mathrm{OVA}}(f(\boldsymbol{x}), y) = \ell\big(g_y(\boldsymbol{x})\big) + \frac{1}{K-1}\sum_{y' \neq y} \ell\big(-g_{y'}(\boldsymbol{x})\big), \tag{3}$$

$$\mathcal{L}_{\mathrm{PC}}\big(f(\boldsymbol{x}), y\big) = \sum_{y' \neq y} \ell\big(g_y(\boldsymbol{x}) - g_{y'}(\boldsymbol{x})\big). \tag{4}$$

Finally, the expectation over unknown $p(\boldsymbol{x}, y)$ in Eq.(1) is empirically approximated using training samples to give a practical classification formulation.

## 3  Classification from complementary labels

In this section, we formulate the problem of complementary classification and propose a risk minimization framework.

We consider the situation where, instead of ordinary class label $y$, we are given only *complementary label* $\overline{y}$ which specifies a class that pattern $\boldsymbol{x}$ does *not* belong to. Our goal is to still learn a classifier

that minimizes the classification risk (1), but only from complementarily labeled training samples $\{(\boldsymbol{x}_i, \overline{y}_i)\}_{i=1}^n$. We assume that $\{(\boldsymbol{x}_i, \overline{y}_i)\}_{i=1}^n$ are drawn independently from an unknown probability distribution with density:[2]

$$\overline{p}(\boldsymbol{x}, \overline{y}) = \frac{1}{K-1} \sum_{y \neq \overline{y}} p(\boldsymbol{x}, y). \tag{5}$$

Let us consider a *complementary* loss $\overline{\mathcal{L}}(f(\boldsymbol{x}), \overline{y})$ for a complementarily labeled sample $(\boldsymbol{x}, \overline{y})$. Then we have the following theorem, which allows unbiased estimation of the classification risk from complementarily labeled samples:

**Theorem 1.** *The classification risk (1) can be expressed as*

$$R(f) = (K-1)\mathbb{E}_{\overline{p}(\boldsymbol{x},\overline{y})}\big[\overline{\mathcal{L}}\big(f(\boldsymbol{x}),\overline{y}\big)\big] - M_1 + M_2, \tag{6}$$

*if there exist constants $M_1, M_2 \geq 0$ such that for all $\boldsymbol{x}$ and $y$, the complementary loss satisfies*

$$\sum_{\overline{y}=1}^K \overline{\mathcal{L}}\big(f(\boldsymbol{x}),\overline{y}\big) = M_1 \quad and \quad \overline{\mathcal{L}}\big(f(\boldsymbol{x}),y\big) + \mathcal{L}\big(f(\boldsymbol{x}),y\big) = M_2. \tag{7}$$

*Proof.* According to (5),

$$(K-1)\mathbb{E}_{\overline{p}(\boldsymbol{x},\overline{y})}[\overline{\mathcal{L}}(f(\boldsymbol{x}),\overline{y})] = (K-1)\int \sum_{\overline{y}=1}^K \overline{\mathcal{L}}(f(\boldsymbol{x}),\overline{y})\overline{p}(\boldsymbol{x},\overline{y})\mathrm{d}\boldsymbol{x}$$

$$= (K-1)\int \sum_{\overline{y}=1}^K \overline{\mathcal{L}}(f(\boldsymbol{x}),\overline{y})\left(\frac{1}{K-1}\sum_{y \neq \overline{y}} p(\boldsymbol{x},y)\right)\mathrm{d}\boldsymbol{x} = \int \sum_{y=1}^K \sum_{\overline{y} \neq y} \overline{\mathcal{L}}(f(\boldsymbol{x}),\overline{y})p(\boldsymbol{x},y)\mathrm{d}\boldsymbol{x}$$

$$= \mathbb{E}_{p(\boldsymbol{x},y)}\left[\sum_{\overline{y} \neq y} \overline{\mathcal{L}}(f(\boldsymbol{x}),\overline{y})\right] = \mathbb{E}_{p(\boldsymbol{x},y)}[M_1 - \overline{\mathcal{L}}(f(\boldsymbol{x}),y)] = M_1 - \mathbb{E}_{p(\boldsymbol{x},y)}[\overline{\mathcal{L}}(f(\boldsymbol{x}),y)],$$

where the fifth equality follows from the first constraint in (7). Subsequently,

$$(K-1)\mathbb{E}_{\overline{p}(\boldsymbol{x},\overline{y})}[\overline{\mathcal{L}}(f(\boldsymbol{x}),\overline{y})] - \mathbb{E}_{p(\boldsymbol{x},y)}[\mathcal{L}(f(\boldsymbol{x}),y)] = M_1 - \mathbb{E}_{p(\boldsymbol{x},y)}[\overline{\mathcal{L}}(f(\boldsymbol{x}),y) + \mathcal{L}(f(\boldsymbol{x}),y)]$$

$$= M_1 - \mathbb{E}_{p(\boldsymbol{x},y)}[M_2]$$

$$= M_1 - M_2,$$

where the second equality follows from the second constraint in (7). $\square$

The first constraint in (7) can be regarded as a multi-class loss version of a symmetric constraint that we later use in Theorem 2. The second constraint in (7) means that the smaller $\mathcal{L}$ is, the larger $\overline{\mathcal{L}}$ should be, i.e., if "pattern $\boldsymbol{x}$ belongs to class $y$" is correct, "pattern $\boldsymbol{x}$ does not belong to class $y$" should be incorrect.

With the expression (6), the classification risk (1) can be naively approximated in an unbiased fashion by the sample average as

$$\widehat{R}(f) = \frac{K-1}{n} \sum_{i=1}^n \overline{\mathcal{L}}\big(f(\boldsymbol{x}_i),\overline{y}_i\big) - M_1 + M_2. \tag{8}$$

Let us define the complementary losses corresponding to the OVA loss $\mathcal{L}_{\mathrm{OVA}}(f(\boldsymbol{x}), y)$ and the PC loss $\mathcal{L}_{\mathrm{PC}}\big(f(\boldsymbol{x}), y\big)$ as

$$\overline{\mathcal{L}}_{\mathrm{OVA}}(f(\boldsymbol{x}), \overline{y}) = \frac{1}{K-1} \sum_{y \neq \overline{y}} \ell\big(g_y(\boldsymbol{x})\big) + \ell\big(-g_{\overline{y}}(\boldsymbol{x})\big), \tag{9}$$

$$\overline{\mathcal{L}}_{\mathrm{PC}}\big(f(\boldsymbol{x}), \overline{y}\big) = \sum_{y \neq \overline{y}} \ell\big(g_y(\boldsymbol{x}) - g_{\overline{y}}(\boldsymbol{x})\big). \tag{10}$$

Then we have the following theorem (its proof is given in Appendix A):

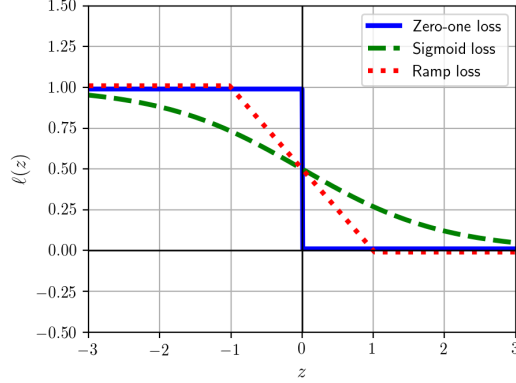

**Figure 1:** Examples of binary losses that satisfy the symmetric condition (11).

**Theorem 2.** *If binary loss $\ell(z)$ satisfies*

$$\ell(z) + \ell(-z) = 1, \tag{11}$$

*then $\overline{\mathcal{L}}_{\mathrm{OVA}}$ satisfies conditions (7) with $M_1 = K$ and $M_2 = 2$, and $\overline{\mathcal{L}}_{\mathrm{PC}}$ satisfies conditions (7) with $M_1 = K(K-1)/2$ and $M_2 = K - 1$.*

For example, the following binary losses satisfy the symmetric condition (11) (see Figure 1):

$$\text{Zero-one loss: } \ell_{0\text{-}1}(z) = \begin{cases} 0 & \text{if } z > 0, \\ 1 & \text{if } z \le 0, \end{cases} \tag{12}$$

$$\text{Sigmoid loss: } \ell_{\mathrm{S}}(z) = \frac{1}{1 + e^z}, \tag{13}$$

$$\text{Ramp loss: } \ell_{\mathrm{R}}(z) = \frac{1}{2} \max\left(0, \min\left(2, 1 - z\right)\right). \tag{14}$$

Note that these losses are non-convex [8]. In practice, the sigmoid loss or ramp loss may be used for training a classifier, while the zero-one loss may be used for tuning hyper-parameters (see Section 6 for the details).

## 4 Estimation Error Bounds

In this section, we establish the estimation error bounds for the proposed method.

Let $\mathcal{G} = \{g(\boldsymbol{x})\}$ be a function class for empirical risk minimization, $\sigma_1, \ldots, \sigma_n$ be $n$ Rademacher variables, then the Rademacher complexity of $\mathcal{G}$ for $\mathcal{X}$ of size $n$ drawn from $p(\boldsymbol{x})$ is defined as follows [23]:

$$\mathfrak{R}_n(\mathcal{G}) = \mathbb{E}_{\mathcal{X}} \mathbb{E}_{\sigma_1, \ldots, \sigma_n} \left[ \sup_{g \in \mathcal{G}} \frac{1}{n} \sum_{\boldsymbol{x}_i \in \mathcal{X}} \sigma_i g(\boldsymbol{x}_i) \right];$$

define the Rademacher complexity of $\mathcal{G}$ for $\overline{\mathcal{X}}$ of size $n$ drawn from $\overline{p}(\boldsymbol{x})$ as

$$\overline{\mathfrak{R}}_n(\mathcal{G}) = \mathbb{E}_{\overline{\mathcal{X}}} \mathbb{E}_{\sigma_1, \ldots, \sigma_n} \left[ \sup_{g \in \mathcal{G}} \frac{1}{n} \sum_{\boldsymbol{x}_i \in \overline{\mathcal{X}}} \sigma_i g(\boldsymbol{x}_i) \right].$$

Note that $\overline{p}(\boldsymbol{x}) = p(\boldsymbol{x})$ and thus $\overline{\mathfrak{R}}_n(\mathcal{G}) = \mathfrak{R}_n(\mathcal{G})$, which enables us to express the obtained theoretical results using the standard Rademacher complexity $\mathfrak{R}_n(\mathcal{G})$.

To begin with, let $\widetilde{\ell}(z) = \ell(z) - \ell(0)$ be the shifted loss such that $\widetilde{\ell}(0) = 0$ (in order to apply the Talagrand's contraction lemma [19] later), and $\widetilde{\mathcal{L}}_{\mathrm{OVA}}$ and $\widetilde{\mathcal{L}}_{\mathrm{PC}}$ be losses defined following (9) and

(10) but with $\widetilde{\ell}$ instead of $\ell$; let $L_\ell$ be any (not necessarily the best) Lipschitz constant of $\ell$. Define the corresponding function classes as follows:

$$\mathcal{H}_{\text{OVA}} = \{(\boldsymbol{x}, \overline{y}) \mapsto \widetilde{\mathcal{L}}_{\text{OVA}}(f(\boldsymbol{x}), \overline{y}) \mid g_1, \ldots, g_K \in \mathcal{G}\},$$
$$\mathcal{H}_{\text{PC}} = \{(\boldsymbol{x}, \overline{y}) \mapsto \widetilde{\mathcal{L}}_{\text{PC}}(f(\boldsymbol{x}), \overline{y}) \mid g_1, \ldots, g_K \in \mathcal{G}\}.$$

Then we can obtain the following lemmas (their proofs are given in Appendices B and C):

**Lemma 3.** *Let $\overline{\mathfrak{R}}_n(\mathcal{H}_{\text{OVA}})$ be the Rademacher complexity of $\mathcal{H}_{\text{OVA}}$ for $\mathcal{S}$ of size $n$ drawn from $\overline{p}(x, \overline{y})$ defined as*

$$\overline{\mathfrak{R}}_n(\mathcal{H}_{\text{OVA}}) = \mathbb{E}_{\mathcal{S}} \mathbb{E}_{\sigma_1, \ldots, \sigma_n} \left[ \sup_{h \in \mathcal{H}_{\text{OVA}}} \frac{1}{n} \sum_{(\boldsymbol{x}_i, \overline{y}_i) \in \mathcal{S}} \sigma_i h(\boldsymbol{x}_i, \overline{y}_i) \right].$$

*Then,*

$$\overline{\mathfrak{R}}_n(\mathcal{H}_{\text{OVA}}) \leq K L_\ell \mathfrak{R}_n(\mathcal{G}).$$

**Lemma 4.** *Let $\overline{\mathfrak{R}}_n(\mathcal{H}_{\text{PC}})$ be the Rademacher complexity of $\mathcal{H}_{\text{PC}}$ defined similarly to $\overline{\mathfrak{R}}_n(\mathcal{H}_{\text{OVA}})$. Then,*

$$\overline{\mathfrak{R}}_n(\mathcal{H}_{\text{PC}}) \leq 2K(K-1) L_\ell \mathfrak{R}_n(\mathcal{G}).$$

Based on Lemmas 3 and 4, we can derive the uniform deviation bounds of $\widehat{R}(f)$ as follows (its proof is given in Appendix D):

**Lemma 5.** *For any $\delta > 0$, with probability at least $1 - \delta$,*

$$\sup_{g_1, \ldots, g_K \in \mathcal{G}} \left| \widehat{R}(f) - R(f) \right| \leq 2K(K-1) L_\ell \mathfrak{R}_n(\mathcal{G}) + (K-1)\sqrt{\frac{2 \ln(2/\delta)}{n}},$$

*where $\widehat{R}(f)$ is w.r.t. $\overline{\mathcal{L}}_{\text{OVA}}$, and*

$$\sup_{g_1, \ldots, g_K \in \mathcal{G}} \left| \widehat{R}(f) - R(f) \right| \leq 4K(K-1)^2 L_\ell \mathfrak{R}_n(\mathcal{G}) + (K-1)^2 \sqrt{\frac{\ln(2/\delta)}{2n}},$$

*where $\widehat{R}(f)$ is w.r.t. $\overline{\mathcal{L}}_{\text{PC}}$.*

Let $(g_1^*, \ldots, g_K^*)$ be the true risk minimizer and $(\widehat{g}_1, \ldots, \widehat{g}_K)$ be the empirical risk minimizer, i.e.,

$$(g_1^*, \ldots, g_K^*) = \arg\min_{g_1, \ldots, g_K \in \mathcal{G}} R(f) \quad \text{and} \quad (\widehat{g}_1, \ldots, \widehat{g}_K) = \arg\min_{g_1, \ldots, g_K \in \mathcal{G}} \widehat{R}(f).$$

Let also

$$f^*(\boldsymbol{x}) = \arg\max_{y \in \{1, \ldots, K\}} g_y^*(\boldsymbol{x}) \quad \text{and} \quad \widehat{f}(\boldsymbol{x}) = \arg\max_{y \in \{1, \ldots, K\}} \widehat{g}_y(\boldsymbol{x}).$$

Finally, based on Lemma 5, we can establish the estimation error bounds as follows:

**Theorem 6.** *For any $\delta > 0$, with probability at least $1 - \delta$,*

$$R(\widehat{f}) - R(f^*) \leq 4K(K-1) L_\ell \mathfrak{R}_n(\mathcal{G}) + (K-1)\sqrt{\frac{8 \ln(2/\delta)}{n}},$$

*if $(\widehat{g}_1, \ldots, \widehat{g}_K)$ is trained by minimizing $\widehat{R}(f)$ is w.r.t. $\overline{\mathcal{L}}_{\text{OVA}}$, and*

$$R(\widehat{f}) - R(f^*) \leq 8K(K-1)^2 L_\ell \mathfrak{R}_n(\mathcal{G}) + (K-1)^2 \sqrt{\frac{2 \ln(2/\delta)}{n}},$$

*if $(\widehat{g}_1, \ldots, \widehat{g}_K)$ is trained by minimizing $\widehat{R}(f)$ is w.r.t. $\overline{\mathcal{L}}_{\text{PC}}$.*

*Proof.* Based on Lemma 5, the estimation error bounds can be proven through

$$R(\widehat{f}) - R(g^*) = \left(\widehat{R}(\widehat{f}) - \widehat{R}(f^*)\right) + \left(R(\widehat{f}) - \widehat{R}(\widehat{f})\right) + \left(\widehat{R}(f^*) - R(f^*)\right)$$

$$\leq 0 + 2 \sup_{g_1,\ldots,g_K \in \mathcal{G}} \left|\widehat{R}(f) - R(f)\right|,$$

where we used that $\widehat{R}(\widehat{f}) \leq \widehat{R}(f^*)$ by the definition of $\widehat{f}$. $\qquad\square$

Theorem 6 also guarantees that learning from complementary labels is consistent: as $n \to \infty$, $R(\widehat{f}) \to R(f^*)$. Consider a linear-in-parameter model defined by

$$\mathcal{G} = \{g(\boldsymbol{x}) = \langle w, \phi(\boldsymbol{x})\rangle_{\mathcal{H}} \mid \|w\|_{\mathcal{H}} \leq C_w, \|\phi(\boldsymbol{x})\|_{\mathcal{H}} \leq C_\phi\},$$

where $\mathcal{H}$ is a Hilbert space with an inner product $\langle\cdot,\cdot\rangle_{\mathcal{H}}$, $w \in \mathcal{H}$ is a normal, $\phi : \mathbb{R}^d \to \mathcal{H}$ is a feature map, and $C_w > 0$ and $C_\phi > 0$ are constants [29]. It is known that $\mathfrak{R}_n(\mathcal{G}) \leq C_w C_\phi/\sqrt{n}$ [23] and thus $R(\widehat{f}) \to R(f^*)$ in $\mathcal{O}_p(1/\sqrt{n})$ if this $\mathcal{G}$ is used, where $\mathcal{O}_p$ denotes the order in probability. This order is already the optimal parametric rate and cannot be improved without additional strong assumptions on $\overline{p}(\boldsymbol{x}, \overline{y})$, $\ell$ and $\mathcal{G}$ jointly.

# 5 Incorporation of ordinary labels

In many practical situations, we may also have ordinarily labeled data in addition to complementarily labeled data. In such cases, we want to leverage both kinds of labeled data to obtain more accurate classifiers. To this end, motivated by [28], let us consider a convex combination of the classification risks derived from ordinarily labeled data and complementarily labeled data:

$$R(f) = \alpha\mathbb{E}_{p(\boldsymbol{x},y)}[\mathcal{L}(f(\boldsymbol{x}), y)] + (1 - \alpha)\Big[(K - 1)\mathbb{E}_{\overline{p}(\boldsymbol{x},\overline{y})}[\overline{\mathcal{L}}(f(\boldsymbol{x}), \overline{y})] - M_1 + M_2\Big], \quad (15)$$

where $\alpha \in [0, 1]$ is a hyper-parameter that interpolates between the two risks. The combined risk (15) can be naively approximated by the sample averages as

$$\widehat{R}(f) = \frac{\alpha}{m}\sum_{j=1}^{m}\mathcal{L}(f(\boldsymbol{x}_j), y_j) + \frac{(1 - \alpha)(K - 1)}{n}\sum_{i=1}^{n}\overline{\mathcal{L}}(f(\boldsymbol{x}_i), \overline{y}_i), \quad (16)$$

where $\{(\boldsymbol{x}_j, y_j)\}_{j=1}^m$ are ordinarily labeled data and $\{(\boldsymbol{x}_i, \overline{y}_i)\}_{i=1}^n$ are complementarily labeled data.

As explained in the introduction, we can naturally obtain both ordinarily and complementarily labeled data through crowdsourcing [14]. Our risk estimator (16) can utilize both kinds of labeled data to obtain better classifiers[3]. We will experimentally demonstrate the usefulness of this combination method in Section 6.

# 6 Experiments

In this section, we experimentally evaluate the performance of the proposed methods.

## 6.1 Comparison of different losses

Here we first compare the performance among four variations of the proposed method with different loss functions: OVA (9) and PC (10), each with the sigmoid loss (13) and ramp loss (14). We used the MNIST hand-written digit dataset, downloaded from the website of the late Sam Roweis[4] (with all patterns standardized to have zero mean and unit variance), with different number of classes: 3 classes (digits "1" to "3") to 10 classes (digits "1" to "9" and "0"). From each class, we randomly sampled 500 data for training and 500 data for testing, and generated complementary labels by randomly selecting one of the complementary classes. From the training dataset, we left out 25% of the data for validating hyperparameter based on (8) with the zero-one loss plugged in (9) or (10).

**Table 1:** Means and standard deviations of classification accuracy over five trials in percentage, when the number of classes ("cls") is changed for the MNIST dataset. "PC" is (10), "OVA" is (9), "Sigmoid" is (13), and "Ramp" is (14). Best and equivalent methods (with 5% t-test) are highlighted in boldface.

| Method | 3 cls | 4 cls | 5 cls | 6 cls | 7 cls | 8 cls | 9 cls | 10 cls |
|---|---|---|---|---|---|---|---|---|
| OVA Sigmoid | **95.2** **(0.9)** | **91.4** **(0.5)** | **87.5** **(2.2)** | **82.0** **(1.3)** | **74.5** **(2.9)** | **73.9** **(1.2)** | **63.6** **(4.0)** | **57.2** **(1.6)** |
| OVA Ramp | **95.1** **(0.9)** | **90.8** **(1.0)** | **86.5** **(1.8)** | **79.4** **(2.6)** | **73.9** **(3.9)** | **71.4** **(4.0)** | **66.1** **(2.1)** | **56.1** **(3.6)** |
| PC Sigmoid | **94.9** **(0.5)** | **90.9** **(0.8)** | **88.1** **(1.8)** | **80.3** **(2.5)** | **75.8** **(2.5)** | **72.9** **(3.0)** | **65.0** **(3.5)** | **58.9** **(3.9)** |
| PC Ramp | **94.5** **(0.7)** | **90.8** **(0.5)** | **88.0** **(2.2)** | **81.0** **(2.2)** | **74.0** **(2.3)** | **71.4** **(2.4)** | **69.0** **(2.8)** | **57.3** **(2.0)** |

For all the methods, we used a linear-in-input model $g_k(\boldsymbol{x}) = \boldsymbol{w}_k^\top \boldsymbol{x} + b_k$ as the binary classifier, where $^\top$ denotes the transpose, $\boldsymbol{w}_k \in \mathbb{R}^d$ is the weight parameter, and $b_k \in \mathbb{R}$ is the bias parameter for class $k \in \{1, \ldots, K\}$. We added an $\ell_2$-regularization term, with the regularization parameter chosen from $\{10^{-4}, 10^{-3}, \ldots, 10^4\}$. Adam [15] was used for optimization with 5,000 iterations, with mini-batch size 100. We reported the test accuracy of the model with the best validation score out of all iterations. All experiments were carried out with Chainer [30].

We reported means and standard deviations of the classification accuracy over five trials in Table 1. From the results, we can see that the performance of all four methods deteriorates as the number of classes increases. This is intuitive because supervised information that complementary labels contain becomes weaker with more classes.

The table also shows that there is no significant difference in classification accuracy among the four losses. Since the PC formulation is regarded as a more direct approach for classification [31] (it takes the sign of the difference of the classifiers, instead of the sign of each classifier as in OVA) and the sigmoid loss is smooth, we use PC with the sigmoid loss as a representative of our proposed method in the following experiments.

## 6.2 Benchmark experiments

Next, we compare our proposed method, PC with the sigmoid loss (PC/S), with two baseline methods. The first baseline is one of the state-of-the-art partial label (PL) methods [5] with the squared hinge loss[5]:

$$\ell(z) = (\max(0, 1 - z))^2.$$

The second baseline is a multi-label (ML) method [3], where every complementary label $\overline{y}$ is translated into a negative label for class $\overline{y}$ and positive labels for the other $K - 1$ classes. This yields the following loss:

$$\mathcal{L}_{\mathrm{ML}}(f(\boldsymbol{x}), \overline{y}) = \sum_{y \neq \overline{y}} \ell(g_y(\boldsymbol{x})) + \ell(- g_{\overline{y}}(\boldsymbol{x})),$$

where we used the same sigmoid loss as the proposed method for $\ell$. We used a one-hidden-layer neural network ($d$-3-1) with *rectified linear units* (ReLU) [24] as activation functions, and weight decay candidates were chosen from $\{10^{-7}, 10^{-4}, 10^{-1}\}$. Standardization, validation and optimization details follow the previous experiments.

We evaluated the classification performance on the following benchmark datasets: WAVEFORM1, WAVEFORM2, SATIMAGE, PENDIGITS, DRIVE, LETTER, and USPS. USPS can be downloaded from the website of the late Sam Roweis[6], and all other datasets can be downloaded from the *UCI machine learning repository*[7]. We tested several different settings of class labels, with equal number of data in each class.

**Table 2:** Means and standard deviations of classification accuracy over 20 trials in percentage. "PC/S" is the proposed method for the pairwise comparison formulation with the sigmoid loss, "PL" is the partial label method with the squared hinge loss, and "ML" is the multi-label method with the sigmoid loss. Best and equivalent methods (with 5% t-test) are highlighted in boldface. "Class" denotes the class labels used for the experiment and "Dim" denotes the dimensionality $d$ of patterns to be classified. "# train" denotes the total number of training and validation samples in each class. "# test" denotes the number of test samples in each class.

| Dataset | Class | Dim | # train | # test | PC/S | PL | ML |
|---------|-------|-----|---------|--------|------|-----|-----|
| WAVEFORM1 | $1 \sim 3$ | 21 | 1226 | 398 | **85.8(0.5)** | **85.7(0.9)** | 79.3(4.8) |
| WAVEFORM2 | $1 \sim 3$ | 40 | 1227 | 408 | **84.7(1.3)** | **84.6(0.8)** | 74.9(5.2) |
| SATIMAGE | $1 \sim 7$ | 36 | 415 | 211 | **68.7(5.4)** | 60.7(3.7) | 33.6(6.2) |
| PENDIGITS | $1 \sim 5$ | 16 | 719 | 336 | **87.0(2.9)** | 76.2(3.3) | 44.7(9.6) |
| | $6 \sim 10$ | | 719 | 335 | **78.4(4.6)** | 71.1(3.3) | 38.4(9.6) |
| | even # | | 719 | 336 | **90.8(2.4)** | 76.8(1.6) | 43.8(5.1) |
| | odd # | | 719 | 335 | **76.0(5.4)** | 67.4(2.6) | 40.2(8.0) |
| | $1 \sim 10$ | | 719 | 335 | **38.0(4.3)** | 33.2(3.8) | 16.1(4.6) |
| DRIVE | $1 \sim 5$ | 48 | 3955 | 1326 | **89.1(4.0)** | 77.7(1.5) | 31.1(3.5) |
| | $6 \sim 10$ | | 3923 | 1313 | **88.8(1.8)** | 78.5(2.6) | 30.4(7.2) |
| | even # | | 3925 | 1283 | **81.8(3.4)** | 63.9(1.8) | 29.7(6.3) |
| | odd # | | 3939 | 1278 | **85.4(4.2)** | 74.9(3.2) | 27.6(5.8) |
| | $1 \sim 10$ | | 3925 | 1269 | **40.8(4.3)** | 32.0(4.1) | 12.7(3.1) |
| LETTER | $1 \sim 5$ | 16 | 565 | 171 | **79.7(5.3)** | **75.1(4.4)** | 28.3(10.4) |
| | $6 \sim 10$ | | 550 | 178 | **76.2(6.2)** | 66.8(2.5) | 34.0(6.9) |
| | $11 \sim 15$ | | 556 | 177 | **78.3(4.1)** | 67.4(3.3) | 28.6(5.0) |
| | $16 \sim 20$ | | 550 | 184 | **77.2(3.2)** | 68.4(2.1) | 32.7(6.4) |
| | $21 \sim 25$ | | 585 | 167 | **80.4(4.2)** | 75.1(1.9) | 32.0(5.7) |
| | $1 \sim 25$ | | 550 | 167 | **5.1(2.1)** | **5.0(1.0)** | **5.2(1.1)** |
| USPS | $1 \sim 5$ | 256 | 652 | 166 | **79.1(3.1)** | 70.3(3.2) | 44.4(8.9) |
| | $6 \sim 10$ | | 542 | 147 | **69.5(6.5)** | **66.1(2.4)** | 37.3(8.8) |
| | even # | | 556 | 147 | **67.4(5.4)** | **66.2(2.3)** | 35.7(6.6) |
| | odd # | | 542 | 147 | **77.5(4.5)** | 69.3(3.1) | 36.6(7.5) |
| | $1 \sim 10$ | | 542 | 127 | **30.7(4.4)** | 26.0(3.5) | 13.3(5.4) |

In Table 2, we summarized the specification of the datasets and reported the means and standard deviations of the classification accuracy over 10 trials. From the results, we can see that the proposed method is either comparable to or better than the baseline methods on many of the datasets.

## 6.3 Combination of ordinary and complementary labels

Finally, we demonstrate the usefulness of combining ordinarily and complementarily labeled data. We used (16), with hyperparameter $\alpha$ fixed at $1/2$ for simplicity. We divided our training dataset by $1 : (K-1)$ ratio, where one subset was labeled ordinarily while the other was labeled complementarily[8]. From the training dataset, we left out 25% of the data for validating hyperparameters based on the zero-one loss version of (16). Other details such as standardization, the model and optimization, and weight-decay candidates follow the previous experiments.

We compared three methods: the ordinary label (OL) method corresponding to $\alpha = 1$, the complementary label (CL) method corresponding to $\alpha = 0$, and the combination (OL & CL) method with $\alpha = 1/2$. The PC and sigmoid losses were commonly used for all methods.

We reported the means and standard deviations of the classification accuracy over 10 trials in Table 3. From the results, we can see that OL & CL tends to outperform OL and CL, demonstrating the usefulnesses of combining ordinarily and complementarily labeled data.

**Table 3:** Means and standard deviations of classification accuracy over 10 trials in percentage. "OL" is the ordinary label method, "CL" is the complementary label method, and "OL & CL" is a combination method that uses both ordinarily and complementarily labeled data. Best and equivalent methods are highlighted in boldface. "Class" denotes the class labels used for the experiment and "Dim" denotes the dimensionality $d$ of patterns to be classified. # train denotes the number of ordinarily/complementarily labeled data for training and validation in each class. # test denotes the number of test data in each class.

| Dataset | Class | Dim | # train | # test | OL ($\alpha = 1$) | CL ($\alpha = 0$) | OL & CL ($\alpha = \frac{1}{2}$) |
|---|---|---|---|---|---|---|---|
| WAVEFORM1 | $1 \sim 3$ | 21 | 413/826 | 408 | 85.3(0.8) | 86.0(0.4) | **86.9(0.5)** |
| WAVEFORM2 | $1 \sim 3$ | 40 | 411/821 | 411 | 82.7(1.3) | 82.0(1.7) | **84.7(0.6)** |
| SATIMAGE | $1 \sim 7$ | 36 | 69/346 | 211 | 74.9(4.9) | 70.1(5.6) | **81.2(1.1)** |
| PENDIGITS | $1 \sim 5$ | 16 | 144/575 | 336 | **91.3(2.1)** | 84.7(3.2) | **93.1(2.0)** |
| | $6 \sim 10$ | | 144/575 | 335 | **86.3(3.5)** | 78.3(6.2) | **87.8(2.8)** |
| | even # | | 144/575 | 336 | 94.3(1.7) | 91.0(4.3) | **95.8(0.6)** |
| | odd # | | 144/575 | 335 | **85.6(2.0)** | 75.9(3.1) | **86.9(1.1)** |
| | $1 \sim 10$ | | 72/647 | 335 | 61.7(4.3) | 41.1(5.7) | **66.9(2.0)** |
| DRIVE | $1 \sim 5$ | 48 | 780/3121 | 1305 | 92.1(2.6) | 89.0(2.1) | **94.2(1.0)** |
| | $6 \sim 10$ | | 795/3180 | 1290 | **87.0(3.0)** | 86.5(3.1) | **89.5(2.1)** |
| | even # | | 657/3284 | 1314 | **91.4(2.9)** | 81.8(4.6) | **91.8(3.3)** |
| | odd # | | 790/3161 | 1255 | 91.1(1.5) | 86.7(2.9) | **93.4(0.5)** |
| | $1 \sim 10$ | | 397/3570 | 1292 | **75.2(2.8)** | 40.5(7.2) | **77.6(2.2)** |
| LETTER | $1 \sim 5$ | 16 | 113/452 | 171 | 85.2(1.3) | 77.2(6.1) | **89.5(1.6)** |
| | $6 \sim 10$ | | 110/440 | 178 | 81.0(1.7) | 77.6(3.7) | **84.6(1.0)** |
| | $11 \sim 15$ | | 111/445 | 177 | 81.1(2.7) | 76.0(3.2) | **87.3(1.6)** |
| | $16 \sim 20$ | | 110/440 | 184 | 81.3(1.8) | 77.9(3.1) | **84.7(2.0)** |
| | $21 \sim 25$ | | 117/468 | 167 | 86.8(2.7) | 81.2(3.4) | **91.1(1.0)** |
| | $1 \sim 25$ | | 22/528 | 167 | 11.9(1.7) | 6.5(1.7) | **31.0(1.7)** |
| USPS | $1 \sim 5$ | 256 | 130/522 | 166 | 83.8(1.7) | 76.5(5.3) | **89.5(1.3)** |
| | $6 \sim 10$ | | 108/434 | 147 | 79.2(2.1) | 67.6(4.3) | **85.5(2.4)** |
| | even # | | 108/434 | 166 | 79.6(2.7) | 67.4(4.4) | **84.8(1.4)** |
| | odd # | | 111/445 | 147 | 82.7(1.9) | 72.9(6.2) | **87.3(2.2)** |
| | $1 \sim 10$ | | 54/488 | 147 | 43.7(2.6) | 28.5(3.6) | **59.3(2.2)** |

# 7 Conclusions

We proposed a novel problem setting called *learning from complementary labels*, and showed that an unbiased estimator to the classification risk can be obtained only from complementarily labeled data, if the loss function satisfies a certain symmetric condition. Our risk estimator can easily be minimized by any stochastic optimization algorithms such as Adam [15], allowing large-scale training. We theoretically established estimation error bounds for the proposed method, and proved that the proposed method achieves the optimal parametric rate. We further showed that our proposed complementary classification can be easily combined with ordinary classification. Finally, we experimentally demonstrated the usefulness of the proposed methods.

The formulation of learning from complementary labels may also be useful in the context of *privacy-aware machine learning* [10]: a subject needs to answer private questions such as psychological counseling which can make him/her hesitate to answer directly. In such a situation, providing a complementary label, i.e., one of the incorrect answers to the question, would be mentally less demanding. We will investigate this issue in the future.

It is noteworthy that the symmetric condition (11), which the loss should satisfy in our complementary classification framework, also appears in other weakly supervised learning formulations, e.g., in positive-unlabeled learning [8]. It would be interesting to more closely investigate the role of this symmetric condition to gain further insight into these different weakly supervised learning problems.

**Acknowledgements**

GN and MS were supported by JST CREST JPMJCR1403. We thank Ikko Yamane for the helpful discussions.

## Footnotes

[1]We normalize the "rest" loss by $K-1$ to be consistent with the discussion in the following sections.

[2]The coefficient $1/(K-1)$ is for the normalization purpose: it would be natural to assume $\overline{p}(\boldsymbol{x}, \overline{y}) = (1/Z) \sum_{y \neq \overline{y}} p(\boldsymbol{x}, y)$ since all $p(\boldsymbol{x}, y)$ for $y \neq \overline{y}$ equally contribute to $\overline{p}(\boldsymbol{x}, \overline{y})$; in order to ensure that $\overline{p}(\boldsymbol{x}, \overline{y})$ is a valid joint density such that $\mathbb{E}_{\overline{p}(\boldsymbol{x},\overline{y})}[1] = 1$, we must take $Z = K - 1$.

[3] Note that when pattern $\boldsymbol{x}$ has already been equipped with ordinary label $y$, giving complementary label $\overline{y}$ does not bring us any additional information (unless the ordinary label is noisy).

[4] See http://cs.nyu.edu/~roweis/data.html.

[5]We decided to use the squared hinge loss (which is convex) here since it was reported to work well in the original paper [5].

[6]See http://cs.nyu.edu/~roweis/data.html.

[7]See http://archive.ics.uci.edu/ml/.

[8]We used $K-1$ times more complementarily labeled data than ordinarily labeled data since a single ordinary label corresponds to $(K-1)$ complementary labels.

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
