[Supplementary Material]

# A Proof of Theorem 2

From Eq.(11), we have

$$\sum_{\overline{y}=1}^{K}\overline{\mathcal{L}}_{\mathrm{OVA}}(f(\boldsymbol{x}),\overline{y}) = \frac{1}{K-1}\sum_{\overline{y}=1}^{K}\sum_{y\neq\overline{y}}\ell\big(g_y(\boldsymbol{x})\big) + \sum_{\overline{y}=1}^{K}\ell\big(-g_{\overline{y}}(\boldsymbol{x})\big)$$

$$= \sum_{\overline{y}=1}^{K}\big(\ell\big(g_{\overline{y}}(\boldsymbol{x})\big) + \ell\big(-g_{\overline{y}}(\boldsymbol{x})\big)\big) = K.$$

$$\mathcal{L}_{\mathrm{OVA}}(f(\boldsymbol{x}),y) + \overline{\mathcal{L}}_{\mathrm{OVA}}(f(\boldsymbol{x}),y) = \ell\big(g_y(\boldsymbol{x})\big) + \frac{1}{K-1}\sum_{\overline{y}\neq y}\ell\big(-g_{\overline{y}}(\boldsymbol{x})\big)$$

$$+ \frac{1}{K-1}\sum_{y'\neq y}\ell\big(g_{y'}(\boldsymbol{x})\big) + \ell\big(-g_y(\boldsymbol{x})\big) = 2,$$

$$\sum_{\overline{y}=1}^{K}\overline{\mathcal{L}}_{\mathrm{PC}}\big(f(\boldsymbol{x}),\overline{y}\big) = \sum_{\overline{y}=1}^{K}\sum_{y\neq\overline{y}}\ell\big(g_y(\boldsymbol{x}) - g_{\overline{y}}(\boldsymbol{x})\big)$$

$$= \sum_{\overline{y}=1}^{K-1}\sum_{y=\overline{y}+1}^{K}\Big(\ell\big(g_y(\boldsymbol{x}) - g_{\overline{y}}(\boldsymbol{x})\big) + \ell\big(g_{\overline{y}}(\boldsymbol{x}) - g_y(\boldsymbol{x})\big)\Big) = \frac{K(K-1)}{2},$$

$$\mathcal{L}_{\mathrm{PC}}(f(\boldsymbol{x}),y) + \overline{\mathcal{L}}_{\mathrm{PC}}(f(\boldsymbol{x}),y) = \sum_{y'\neq y}\ell\big(g_y(\boldsymbol{x}) - g_{y'}(\boldsymbol{x})\big) + \sum_{y'\neq y}\ell\big(g_{y'}(\boldsymbol{x}) - g_y(\boldsymbol{x})\big) = K-1.$$

$$\square$$

# B Proof of Lemma 3

By definition, $h(\boldsymbol{x}_i,\overline{y}_i) = \widetilde{\mathcal{L}}_{\mathrm{OVA}}(f(\boldsymbol{x}_i),\overline{y}_i)$ so that

$$\overline{\mathfrak{R}}_n(\mathcal{H}_{\mathrm{OVA}}) = \mathbb{E}_{\mathcal{S}}\mathbb{E}_\sigma\left[\sup_{g_1,\ldots,g_K\in\mathcal{G}}\frac{1}{n}\sum_{(\boldsymbol{x}_i,\overline{y}_i)\in\mathcal{S}}\sigma_i\left(\frac{1}{K-1}\sum_{y\neq\overline{y}}\widetilde{\ell}(g_y(\boldsymbol{x}_i)) + \widetilde{\ell}(-g_{\overline{y}_i}(\boldsymbol{x}_i))\right)\right].$$

After rewriting $\widetilde{\mathcal{L}}_{\mathrm{OVA}}(f(\boldsymbol{x}_i),\overline{y}_i)$, we can know that

$$\widetilde{\mathcal{L}}_{\mathrm{OVA}}(f(\boldsymbol{x}_i),\overline{y}_i) = \frac{1}{K-1}\sum_{y}\widetilde{\ell}(g_y(\boldsymbol{x}_i)) + \frac{K-2}{K-1}\widetilde{\ell}(-g_{\overline{y}_i}(\boldsymbol{x}_i)),$$

and subsequently,

$$\overline{\mathfrak{R}}_n(\mathcal{H}_{\mathrm{OVA}}) \leq \frac{1}{K-1}\mathbb{E}_{\mathcal{S}}\mathbb{E}_\sigma\left[\sup_{g_1,\ldots,g_K\in\mathcal{G}}\frac{1}{n}\sum_{(\boldsymbol{x}_i,\overline{y}_i)\in\mathcal{S}}\sigma_i\sum_y\widetilde{\ell}(g_y(\boldsymbol{x}_i))\right]$$

$$+ \frac{K-2}{K-1}\mathbb{E}_{\mathcal{S}}\mathbb{E}_\sigma\left[\sup_{g_1,\ldots,g_K\in\mathcal{G}}\frac{1}{n}\sum_{(\boldsymbol{x}_i,\overline{y}_i)\in\mathcal{S}}\sigma_i\widetilde{\ell}(-g_{\overline{y}_i}(\boldsymbol{x}_i))\right]$$

due to the sub-additivity of the supremum.

The first term is independent of $\overline{y}_i$ and thus

$$\mathbb{E}_{\mathcal{S}}\mathbb{E}_{\sigma}\left[\sup_{g_1,\ldots,g_K\in\mathcal{G}}\frac{1}{n}\sum_{(\boldsymbol{x}_i,\overline{y}_i)\in\mathcal{S}}\sigma_i\sum_y\widetilde{\ell}(g_y(\boldsymbol{x}_i))\right] = \mathbb{E}_{\overline{\mathcal{X}}}\mathbb{E}_{\sigma}\left[\sup_{g_1,\ldots,g_K\in\mathcal{G}}\frac{1}{n}\sum_{\boldsymbol{x}_i\in\overline{\mathcal{X}}}\sigma_i\sum_y\widetilde{\ell}(g_y(\boldsymbol{x}_i))\right]$$

$$\leq \sum_y\mathbb{E}_{\overline{\mathcal{X}}}\mathbb{E}_{\sigma}\left[\sup_{g_1,\ldots,g_K\in\mathcal{G}}\frac{1}{n}\sum_{\boldsymbol{x}_i\in\overline{\mathcal{X}}}\sigma_i\widetilde{\ell}(g_y(\boldsymbol{x}_i))\right]$$

$$= \sum_y\mathbb{E}_{\overline{\mathcal{X}}}\mathbb{E}_{\sigma}\left[\sup_{g_y\in\mathcal{G}}\frac{1}{n}\sum_{\boldsymbol{x}_i\in\overline{\mathcal{X}}}\sigma_i\widetilde{\ell}(g_y(\boldsymbol{x}_i))\right]$$

$$= K\overline{\mathfrak{R}}_n(\widetilde{\ell}\circ\mathcal{G})$$

which means the first term can be bounded by $K/(K-1)\cdot\overline{\mathfrak{R}}_n(\widetilde{\ell}\circ\mathcal{G})$. The second term is more involved. Let $I(\cdot)$ be the indicator function and $\alpha_i = 2I(y=\overline{y}_i)-1$, then

$$\mathbb{E}_{\mathcal{S}}\mathbb{E}_{\sigma}\left[\sup_{g_1,\ldots,g_K\in\mathcal{G}}\frac{1}{n}\sum_{(\boldsymbol{x}_i,\overline{y}_i)\in\mathcal{S}}\sigma_i\widetilde{\ell}(-g_{\overline{y}_i}(\boldsymbol{x}_i))\right]$$

$$= \mathbb{E}_{\mathcal{S}}\mathbb{E}_{\sigma}\left[\sup_{g_1,\ldots,g_K\in\mathcal{G}}\frac{1}{n}\sum_{(\boldsymbol{x}_i,\overline{y}_i)\in\mathcal{S}}\sigma_i\sum_y\widetilde{\ell}(-g_y(\boldsymbol{x}_i))I(y=\overline{y}_i)\right]$$

$$= \mathbb{E}_{\mathcal{S}}\mathbb{E}_{\sigma}\left[\sup_{g_1,\ldots,g_K\in\mathcal{G}}\frac{1}{2n}\sum_{(\boldsymbol{x}_i,\overline{y}_i)\in\mathcal{S}}\sigma_i\sum_y\widetilde{\ell}(-g_y(\boldsymbol{x}_i))(\alpha_i+1)\right]$$

$$\leq \mathbb{E}_{\mathcal{S}}\mathbb{E}_{\sigma}\left[\sup_{g_1,\ldots,g_K\in\mathcal{G}}\frac{1}{2n}\sum_{(\boldsymbol{x}_i,\overline{y}_i)\in\mathcal{S}}\alpha_i\sigma_i\sum_y\widetilde{\ell}(-g_y(\boldsymbol{x}_i))\right]$$

$$+ \mathbb{E}_{\mathcal{S}}\mathbb{E}_{\sigma}\left[\sup_{g_1,\ldots,g_K\in\mathcal{G}}\frac{1}{2n}\sum_{(\boldsymbol{x}_i,\overline{y}_i)\in\mathcal{S}}\sigma_i\sum_y\widetilde{\ell}(-g_y(\boldsymbol{x}_i))\right]$$

$$= \mathbb{E}_{\mathcal{S}}\mathbb{E}_{\sigma}\left[\sup_{g_1,\ldots,g_K\in\mathcal{G}}\frac{1}{n}\sum_{(\boldsymbol{x}_i,\overline{y}_i)\in\mathcal{S}}\sigma_i\sum_y\widetilde{\ell}(-g_y(\boldsymbol{x}_i))\right],$$

where we used that $\alpha_i\sigma_i$ has exactly the same distribution as $\sigma_i$. This can be similarly bounded by $\overline{\mathfrak{R}}_n(\widetilde{\ell}\circ\mathcal{G})$ and the second term can be bounded by $K(K-2)/(K-1)\cdot\overline{\mathfrak{R}}_n(\widetilde{\ell}\circ\mathcal{G})$.

As a result,

$$\overline{\mathfrak{R}}_n(\mathcal{H}_{\mathrm{OVA}}) \leq \left(\frac{K}{K-1}+\frac{K(K-2)}{K-1}\right)\overline{\mathfrak{R}}_n(\widetilde{\ell}\circ\mathcal{G})$$

$$= K\overline{\mathfrak{R}}_n(\widetilde{\ell}\circ\mathcal{G})$$

$$\leq KL_\ell\overline{\mathfrak{R}}_n(\mathcal{G})$$

$$= KL_\ell\mathfrak{R}_n(\mathcal{G}),$$

according to Talagrand's contraction lemma [19]. $\qquad\square$

## C  Proof of Lemma 4

By definition,

$$\overline{\mathfrak{R}}_n(\mathcal{H}_{\mathrm{PC}}) = \mathbb{E}_{\mathcal{S}}\mathbb{E}_{\sigma}\left[\sup_{g_1,\ldots,g_K\in\mathcal{G}}\frac{1}{n}\sum_{(\boldsymbol{x}_i,\overline{y}_i)\in\mathcal{S}}\sigma_i\left(\sum_{y'\neq\overline{y}_i}\widetilde{\ell}(g_{y'}(\boldsymbol{x}_i)-g_{\overline{y}_i}(\boldsymbol{x}_i))\right)\right].$$

Using the proof technique for handling the second term in the proof of Lemma 3, we have

$$\overline{\mathfrak{R}}_n(\mathcal{H}_{\mathrm{PC}}) \leq \mathbb{E}_{\mathcal{S}}\mathbb{E}_{\sigma}\left[\sup_{g_1,\dots,g_K\in\mathcal{G}}\frac{1}{n}\sum_{(\boldsymbol{x}_i,\overline{y}_i)\in\mathcal{S}}\sigma_i\sum_{y}\left(\sum_{y'\neq y}\widetilde{\ell}(g_{y'}(\boldsymbol{x}_i)-g_y(\boldsymbol{x}_i))\right)\right]$$

$$= \mathbb{E}_{\overline{\mathcal{X}}}\mathbb{E}_{\sigma}\left[\sup_{g_1,\dots,g_K\in\mathcal{G}}\frac{1}{n}\sum_{\boldsymbol{x}_i\in\overline{\mathcal{X}}}\sigma_i\sum_{y}\left(\sum_{y'\neq y}\widetilde{\ell}(g_{y'}(\boldsymbol{x}_i)-g_y(\boldsymbol{x}_i))\right)\right]$$

$$\leq \sum_{y}\sum_{y'\neq y}\mathbb{E}_{\overline{\mathcal{X}}}\mathbb{E}_{\sigma}\left[\sup_{g_y,g_{y'}\in\mathcal{G}}\frac{1}{n}\sum_{\boldsymbol{x}_i\in\overline{\mathcal{X}}}\sigma_i\widetilde{\ell}(g_{y'}(\boldsymbol{x}_i)-g_y(\boldsymbol{x}_i))\right],$$

due to the sub-additivity of the supremum.

Let
$$\mathcal{G}_{y,y'} = \{\boldsymbol{x}\mapsto g_{y'}(\boldsymbol{x})-g_y(\boldsymbol{x}) \mid g_y,g_{y'}\in\mathcal{G}\},$$
then according to Talagrand's contraction lemma [19],

$$\mathbb{E}_{\overline{\mathcal{X}}}\mathbb{E}_{\sigma}\left[\sup_{g_y,g_{y'}\in\mathcal{G}}\frac{1}{n}\sum_{\boldsymbol{x}_i\in\overline{\mathcal{X}}}\sigma_i\widetilde{\ell}(g_{y'}(\boldsymbol{x}_i)-g_y(\boldsymbol{x}_i))\right]$$

$$= \overline{\mathfrak{R}}_n(\widetilde{\ell}\circ\mathcal{G}_{y,y'})$$

$$\leq L_\ell\overline{\mathfrak{R}}_n(\mathcal{G}_{y,y'})$$

$$= L_\ell\mathbb{E}_{\overline{\mathcal{X}}}\mathbb{E}_{\sigma}\left[\sup_{g_y,g_{y'}\in\mathcal{G}}\frac{1}{n}\sum_{\boldsymbol{x}_i\in\overline{\mathcal{X}}}\sigma_i(g_{y'}(\boldsymbol{x}_i)-g_y(\boldsymbol{x}_i))\right]$$

$$\leq L_\ell\mathbb{E}_{\overline{\mathcal{X}}}\mathbb{E}_{\sigma}\left[\sup_{g_y\in\mathcal{G}}\frac{1}{n}\sum_{\boldsymbol{x}_i\in\overline{\mathcal{X}}}\sigma_ig_y(\boldsymbol{x}_i)\right] + L_\ell\mathbb{E}_{\overline{\mathcal{X}}}\mathbb{E}_{\sigma}\left[\sup_{g_{y'}\in\mathcal{G}}\frac{1}{n}\sum_{\boldsymbol{x}_i\in\overline{\mathcal{X}}}\sigma_ig_{y'}(\boldsymbol{x}_i)\right]$$

$$= 2L_\ell\overline{\mathfrak{R}}_n(\mathcal{G})$$

$$= 2L_\ell\mathfrak{R}_n(\mathcal{G}).$$

This proves that $\overline{\mathfrak{R}}_n(\mathcal{H}_{\mathrm{PC}}) \leq 2K(K-1)L_\ell\mathfrak{R}_n(\mathcal{G})$. $\qquad\square$

## D  Proof of Lemma 5

We are going to prove the case of $\overline{\mathcal{L}}_{\mathrm{OVA}}$; the other case is similar. We consider a single direction $\sup_{g_1,\dots,g_K\in\mathcal{G}}(\widehat{R}(f)-R(f))$ with probability at least $1-\delta/2$; the other direction is similar too.

Given the symmetric condition (11), it must hold that $\|\overline{\mathcal{L}}_{\mathrm{OVA}}\|_\infty = 2$ when $g_1,\dots,g_K$ can be any measurable functions. Let a single $(\boldsymbol{x}_i,\overline{y}_i)$ be replaced with $(\boldsymbol{x}_i',\overline{y}_i')$, then the change of $\sup_{g_1,\dots,g_K\in\mathcal{G}}(\widehat{R}(f)-R(f))$ is no greater than $2(K-1)/n$. Apply McDiarmid's inequality [22] to the single-direction uniform deviation $\sup_{g_1,\dots,g_K\in\mathcal{G}}(\widehat{R}(f)-R(f))$ to get that

$$\Pr\left\{\sup_{g_1,\dots,g_K\in\mathcal{G}}(\widehat{R}(f)-R(f)) - \mathbb{E}\left[\sup_{g_1,\dots,g_K\in\mathcal{G}}(\widehat{R}(f)-R(f))\right] \geq \epsilon\right\} \leq \exp\left(-\frac{2\epsilon^2}{n(2(K-1)/n)^2}\right),$$

or equivalently, with probability at least $1-\delta/2$,

$$\sup_{g_1,\dots,g_K\in\mathcal{G}}(\widehat{R}(f)-R(f)) \leq \mathbb{E}\left[\sup_{g_1,\dots,g_K\in\mathcal{G}}(\widehat{R}(f)-R(f))\right] + (K-1)\sqrt{\frac{2\ln(2/\delta)}{n}}.$$

Since $R(f) = \mathbb{E}[\widehat{R}(f)]$, it is a routine work to show by symmetrization that [23]

$$\mathbb{E}\left[\sup_{g_1,\dots,g_K\in\mathcal{G}}(\widehat{R}(f)-R(f))\right] \leq 2(K-1)\overline{\mathfrak{R}}_n(\mathcal{H}_{\mathrm{OVA}})$$

$$\leq 2K(K-1)L_\ell\mathfrak{R}_n(\mathcal{G}),$$

where the last line is due to Lemma 3. $\qquad\square$