[Reviews · NeurIPS 2017]

Reviewer 1



Collecting labeled data is costly and the paper considers a new setting, where only complementary label that specifies a class where a pattern does not belong to, is collected. It is obviously that the collection of complementary label is much more easy than that of precise label. The paper presents to learn a model that minimizes the loss in terms of the complementary label. The paper then extensively studies the estimation error bound of the proposal, by using unbiased estimator given that a condition of loss function hold. Experiments are conducted on a number of data sets, and results show encouraging performance. The paper is clearly written. The studied problem is interesting and is of importance for several applications. The proposed method is sound, as it is based on statistical learning. The theoretical studies are extensive. To my knowledge, it is new. Experiment results are rather good. Two minor comments: 1) The theoretical analysis shows that the estimator error bound is related to the number of $k$. It would be interesting to see how $k$ affects the empirical performance of the proposal. 2) If the space is not sufficient for new empirical studies, the author could compress the theoretical part and make it more compact, such as, putting some proofs in supplementary files.

Reviewer 2



This paper proposes a novel problem setting and algorithm for learning from complementary labels. It shows an unbiased estimator of the classification risk can be obtained only from complementary labels, if a loss function satisfies a particular symmetric condition. The paper also provides estimation error bounds for the proposed method. Experiments show the approach can produce results similar to standard learning with ordinary labels on some datasets. The idea and the theoretical results are interesting. But I question the usefulness of the proposed learning problem. Though using complementary labels can be less laborious than ordinary labels, complementary labels are less informative than ordinary labels. If using N training instances with complementary labels cannot perform better (this is almost certain) than learning with N/(K-1) training instances with ordinary labels, why should one use complementary labels? Moreover, based on the theoretical results, learning with complementary labels are restricted to a limited number of loss functions, while there are many state-of-the-art effective learning methods with ordinary labels. The paper only suggests in the conclusion section that such complementary label learning can be useful in the context of privacy-aware machine learning. Then why not put this study within the privacy-aware learning setting and compare to the privacy-aware learning techniques? For section 3: (1) On line 51, it says “ that incurs a large loss for large z”. Shouldn’t be “that incurs a large loss for small z”? (2) It is not clear how the derivation on Line 79-80 was conducted. Why should \sum_{y\not= \bar{y}}\ell(g_y(x))/(K-1) = \ell(g_{\bar{y}}(x)) ? The experiments are very basic. In the benchmark experiments, it only compares to a self-constructed ML formulation and a one-hidden-layer neural network with ordinary labels. The way of translating the complementary labels to multi-labels is not proper since it actually brings much wrong information into the labels. The one-hidden-layer neural network is far from being a state-of-the-art multi-label classification model. It is better to just compare with the standard learning with ordinary labels using the same PC with sigmoid loss. Another issue is: why not use the whole datasets instead of part of them? It would also be interesting to see the comparison between complementary label learning with ordinary label learning with a range of different number of classes on the same datasets.